# The Future of Total Elbow Arthroplasty: A Statistical Forecast Model for Germany

**DOI:** 10.3390/healthcare12131322

**Published:** 2024-07-02

**Authors:** Felix Krane, Vincent Johann Heck, Jannik Leyendecker, Kristina Klug, Alexander Klug, Michael Hackl, Jörn Kircher, Lars Peter Müller, Tim Leschinger

**Affiliations:** 1University Hospital Cologne, Department of Orthopedics, Trauma and Plastic Surgery, University of Cologne, Kerpener Str. 62, 50937 Cologne, Germany; 2Department of Psychology, Goethe-University Frankfurt, Theodor-W.-Adorno Platz 6, PEG, 60629 Frankfurt am Main, Germany; 3Department of Trauma and Orthopedic Surgery, BG Unfallklinik Frankfurt am Main, Friedberger Landstr. 430, 60389 Frankfurt am Main, Germany; 4Department of Shoulder and Elbow Surgery, ATOS Klinik Fleetinsel Hamburg, Admiralitätstrasse 3-4, 20459 Hamburg, Germany; 5Medical Faculty, Heinrich-Heine-University Düsseldorf, Moorenstr. 5, 40255 Düsseldorf, Germany

**Keywords:** forecast, total elbow arthroplasty, total arthroplasty, elbow replacement, demographic development

## Abstract

This study provides a statistical forecast for the development of total elbow arthroplasties (TEAs) in Germany until 2045. The authors used an autoregressive integrated moving average (ARIMA), Error-Trend-Seasonality (ETS), and Poisson model to forecast trends in total elbow arthroplasty based on demographic information and official procedure statistics. They predict a significant increase in total elbow joint replacements, with a higher prevalence among women than men. Comprehensive national data provided by the Federal Statistical Office of Germany (Statistisches Bundesamt) were used to quantify TEA’s total number and incidence rates. Poisson regression, exponential smoothing with Error-Trend-Seasonality, and autoregressive integrated moving average models (ARIMA) were used to predict developments in the total number of surgeries until 2045. Overall, the number of TEAs is projected to increase continuously from 2021 to 2045. This will result in a total number of 982 (TEAs) in 2045 of mostly elderly patients above 80 years. Notably, female patients will receive TEAs 7.5 times more often than men. This is likely influenced by demographic and societal factors such as an ageing population, changes in healthcare access and utilization, and advancements in medical technology. Our projection emphasises the necessity for continuous improvements in surgical training, implant development, and rehabilitation protocols.

## 1. Introduction

The baby boomer generation will reach retirement age in the upcoming years, which poses challenges to global healthcare systems and likely increases the incidence of degenerative joint diseases [1,2]. The number of arthroscopies of the lower limb is expected to decrease within the next few years, while joint replacements of the upper extremities are expected to grow [3,4,5,6,7]. Additionally, changes in populations with low birth and low immigration rates are also affecting the demographic structure [8,9,10]. An abundance of scientific projections for the development of performed arthroplasties has been published in recent years. Until now, no predictions have been made for the development of total elbow arthroplasty (TEA) in Germany. TEA has undergone significant changes in recent years, with shifts in indications and advancements in surgical techniques and prosthetic designs. For instance, the most common indication for total elbow arthroplasty has transitioned from inflammatory arthritis to traumatic or post-traumatic pathologies [11,12,13]. Advancements in prosthetic designs have also influenced the landscape of TEA. Some prostheses now incorporate features like anterior flanges to enhance implant stability and peri-articular load transfer, aiming to reduce complications such as distal humerus cortical strains. Modular designs broaden adaption to conditions such as distal humeral fractures or radial head complications. [14,15,16] TEA aims to reduce pain, restore elbow function, and thus improve the overall quality of life of the patients. TEA is commonly used to treat severe fractures, failed osteosynthesis, degenerative joint diseases, rheumatoid arthritis (RA), and tumours [17,18,19,20]. In recent years, the numbers for TEA in RA have decreased due to improved medication [21,22]. By that, the main indication for TEA in Germany today is the acute or post-traumatic situation. Today, surgeons can choose from various models and sizes for different indications, emphasising the relevance of TEA in the following years.

The Autoregressive Integrated Moving Average (ARIMA), Error–Trend–Seasonal (ETS), and Poisson Regression (POR) are statistical models used for time series forecasting based on historical data. ARIMA models consider the correlation between an observation and a lagged observation, differences between observations, and moving averages of previous errors to model the data. ETS models are based on decomposing the data into trend, seasonal, and error components, which are then modelled separately. Poisson models are used when the response variable is a count and follows a Poisson distribution, and the number of occurrences is modelled at a fixed interval. These models have been used in medicine for various applications, such as epidemiology, disease forecasting, and healthcare demand prediction [23,24,25]. This study aimed to forecast the incidence and total number of TEA in Germany in the coming years.

## 2. Materials and Methods

This statistical analysis used comprehensive nationwide data provided by the Federal Statistical Office of Germany (Statistisches Bundesamt, Wiesbaden, Germany). The provided data include inpatient treatment reports from all German hospitals. The data are based on the International Statistical Classification of Diseases and Related Health Problems, Tenth Edition (ICD-10), and the System for Procedures in Medicine (OPS). The data are anonymised, relevant for the billing of the clinics, and subject to official audits by the authorities. To quantify, the total number and incidence rates of TEA depending on the calendar year, age, and sex, the numbers provided were divided into age groups of 10 years. In addition, data regarding the development of the national population in Germany, including age and sex, were retrieved from the official website of the Federal Statistical Office of Germany.

Three different projections were implemented using these data. POR, ETS, and ARIMA were used in the historical procedure rates from 2005 to 2021, which are official population projections from 2020 to 2045. All projections assumed that the observed trend of the past years would continue. Afterwards, the most accurate model was identified by the implementation of the mean absolute percentage error (MAPE), the root mean squared error (RMSE), and the mean absolute error (MAE). The statistical analyses were conducted using R Version 3.4.0 (R Development Core Team, The R Foundation for Statistical Computing, Vienna, Austria).

## 3. Results

Over the past few years, the total number of TEAs has steadily increased. At the same time, the population has become older. Based on exponential smoothing modelling, the number of TEAs is projected to increase from 2021 to 2045 continuously (Figure 1 and Figure 2). Specifically, the incidence rate of TEA is expected to grow by around 63% to 1.2 per 100,000 inhabitants [95% CI 1.1–1.3], leading to a projected total number of 982 TEA [95% CI 882–1060] in 2045 up from 608 in 2021.

The three prediction models used in this study to estimate the future incidence of TEA are ARIMA, ETS, and Poisson. The accuracy of these models was evaluated using three metrics: Mean Absolute Percentage Error (MAPE), Root Mean Squared Error (RMSE), and Mean Absolute Error (MAE). The ETS model showed the highest accuracy among the three models, with a MAPE of 5.8%, RMSE of 37.4, and MAE of 31.6. The ARIMA model had a MAPE of 7.3%, RMSE of 47.9, and MAE of 39.3, while the Poisson model had a MAPE of 8.2%, RMSE of 58.8, and MAE of 51.4 (Table 1). These results suggest that the ETS model is the most accurate for predicting the incidence of TEA in the future. However, the accuracy of these models can vary depending on the specific dataset and context in which they are used. Therefore, it is recommended that the performance of different models be evaluated on a case-by-case basis before selecting a model for future predictions.

Based on our projection, this increase is mainly driven by a rising number of predominantly female patients aged 80 years and older. This is expected to lead to a 2.3-fold increase in performed procedures for all patients aged 80 years and older in 2045 compared with 2021. In total, TEA will be performed seven times more often in women than in men in 2045, corresponding to a projected total number of 866 TEA [95% CI 785–948] for women and 116 [95% CI 86–146] for men in 2045.

The data in the table show the projected total numbers and incidence rates of total elbow arthroplasty (TEA) in Germany from 2021 to 2045 (Table 2). The data are based on comprehensive nationwide data provided by the Federal Statistical Office of Germany and include information on TEA’s total number and incidence rates as a function of calendar year, age, and gender. The population is expected to increase from 82.7 million in 2021 to 83.76 million in 2024 and will then slightly decline annually until it reaches 82.1 million in 2045. This increase in population, in combination with increasing cases and an overall ageing population, is reflected in the projected total number of TEA procedures, which is expected to rise from 608 in 2021 to 982 in 2045.

The data also show differences in the incidence rates of TEA between males and females. In 2021, the incidence rate of TEA was 0.74 per 100,000 individuals, with a projected relative increase of 0.94% in males and 0.73% in females in 2022. This trend is anticipated to persist, with the incidence rate of TEA projected to rise faster in males than in females until 2031. After 2031, the incidence rate is expected to increase more quickly in females than in males (Figure 3).

Furthermore, the data provide information on the projected total number and incidence rates of TEA as a function of age. The data show that TEA is performed most often in individuals aged 70 years and older, with the incidence rate increasing steadily (Figure 4). The highest incidence rate is projected for individuals aged 80 years and older.

## 4. Discussion

Statistical models such as ARIMA, ETS, and Poisson are used to make predictions for future counts of surgeries based on historical trends. The provided data show the incidence rates of TEA for men and women from 2005 to 2045. Throughout the years, there have been notable sex differences in the incidence rates according to the official numbers provided by the government. In 2005, the incidence rate for TEA in men was 0.18, while for women, it was higher at 0.43. Over the years, both incidence rates generally increased, but women consistently have higher rates than men. In 2020, the incidence rate for men was 0.27, while for women, it was 1.08. This trend continues until 2045, with women consistently having higher incidence rates than men. The abovementioned differences in incidence rates suggest variations in susceptibility, risk factors, or exposure to the condition between men and women. Investigating potential underlying factors contributing to these discrepancies is essential to understand the condition better and develop targeted prevention and treatment strategies for both sexes. Our projection data indicate that the observed sex differences in the increase in TEA cases until 2045 are primarily driven by a rising number of patients who are 80 years and older. This is further compounded by the fact that women generally have a more significant life expectancy than men, which partially explains the sex differences in projected cases for TEA. This is highlighted by the overall increase in such procedures for patients 80 and older, which will be 2.3-fold when 2045 data are compared with 2021 as the base year. However, other factors such as different prevalence and severity of elbow joint disorders between men and women or sex-dependent healthcare-seeking behaviour can cause this discrepancy between sexes [26,27]. Furthermore, different bone conditions caused by osteoporosis impact the prevalence of complex fractures of the forearm [28].

Orthopaedic surgeons in Germany face unique challenges due to demographic changes. Factors contributing to these demographic shifts include low birth rates and limited immigration, resulting in a declining population [29]. With an ageing population and increasing life expectancy, the prevalence of orthopaedic pathologies such as femoral neck fractures and the need for hip, shoulder, and knee replacements are expected to improve continuously [24,30].

Traumatic conditions substantially influenced the development of TEA within the last decade. While inflammatory diseases remain the main indication for TEA, acute trauma represents the second most prevalent indication, with increasing numbers. This is highlighted by an increase in severe injuries like AO Type C fractures over the past two decades [31]. This is likely associated with ageing, as mentioned above in (Western) societies [32]. Recent studies have shown the superiority of treatments with TEA for complex fractures such as Dubberly Type B or multifragmentary fractures of the trochlea without the possibility of osteosynthesis [33,34,35]. Given the continuous development of implants, it is to be expected that complex fractures in elderly patients will be treated with TEA in the future.

Simultaneously, the percentage of TEA indications based on inflammatory joint diseases is decreasing, most likely due to innovations in anti-inflammatory medications such as disease-modifying drugs (DMARD) and antibodies in industrialised countries [36,37]. Nevertheless, a relevant number of patients suffering from RA require TEA, as evidenced by virtually unchanging incidences in this population [38,39]. However, our projection suggests a trend towards trauma as the main indication for TEA. Still, it might be influenced by the increasing number of rheumatic joint diseases in the overall population [36].

Additionally, the increase in primary TEA will inevitably trigger an increase in complications, such as infections, periprosthetic fractures, and mechanical failure. Consequentially, revision TEA will pose an additional challenge for the surgical team. This emphasises the necessity for more research and innovations to improve diagnostics for infections and treatments for periprosthetic fractures or loosening, as well as overall longevity.

## 5. Conclusions

Predicting future technological breakthroughs is challenging [40]. However, ongoing research and development in elbow prostheses suggest potential advances. For example, advancements in biomaterials may lead to new implant materials with improved mechanical properties and biocompatibility [41,42,43]. Those might result in longer-lasting, more durable, and personalised prostheses [44,45]. Similarly, developing new surgical techniques and technologies, such as robotics and virtual reality, may enhance the accuracy and precision of implant placement and reduce the risk of complications [46,47].

Fractures in elderly patients are common. The demographic changes will likely significantly increase the total number of fractures, leading to an increased need for total arthroplasties in the upper limb, including the elbow [7,48]. A high activity level for TEA might aggravate the problem, causing more fracture-related indications [49]. On the other hand, the increasing demand for joint replacement surgery may strain healthcare resources and lead to longer wait times for surgery. Elbow arthroplasty is a technically challenging procedure that requires a high level of skill and experience, which can limit its availability in certain regions or healthcare systems. More surgeons need to be trained in surgical techniques for TEA. Therefore, ongoing TEA research, training, and development are essential to ensure patients receive the best possible care.

While short-term outcomes of TEA surgery are favourable, there is a lack of data regarding elbow prostheses’ long-term effectiveness and durability [50,51]. This is particularly important because contemporary joint replacement surgery is frequently performed in younger patients who may require revision surgery later in life [52,53].

The authors acknowledge the limitations of the presented manuscript. Most importantly, this work is based on official numbers provided by the German government. The forecasted numbers refer to an industrialised country with an overall high level of medical care. The developments described above for Germany are certainly transferable to other countries. Still, they cannot reflect the trends for regions with different medical systems or demographics and thus lack generalisability. The statistical analysis in this paper provides trend values that can predict trends that can be affected by unpredictable events such as international crises, wars or pandemics. Overall, this work can be used to better plan and manage future challenges in TEA.

## Figures and Tables

**Figure 1 healthcare-12-01322-f001:**
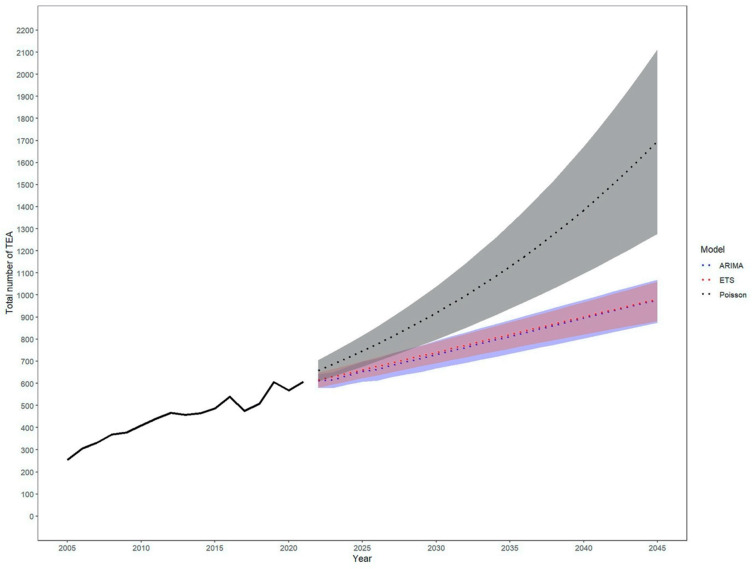
Increase depending on the model chosen (Autoregressive Integrated Moving Average (ARIMA), Error–Trend–Seasonality (ETS), and Poisson). The historical data is displayed by the solid line. The statistical models differ and are shown in the colour-coded section.

**Figure 2 healthcare-12-01322-f002:**
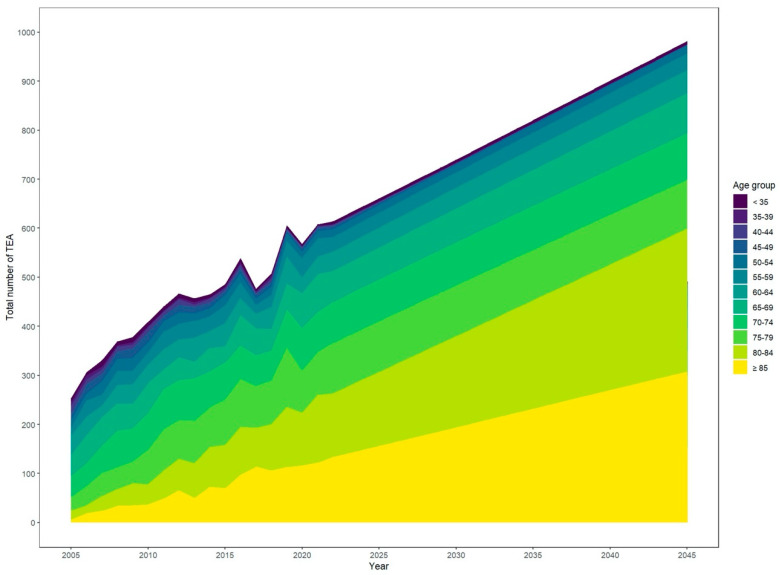
Projected overall increase in total elbow arthroplasty.

**Figure 3 healthcare-12-01322-f003:**
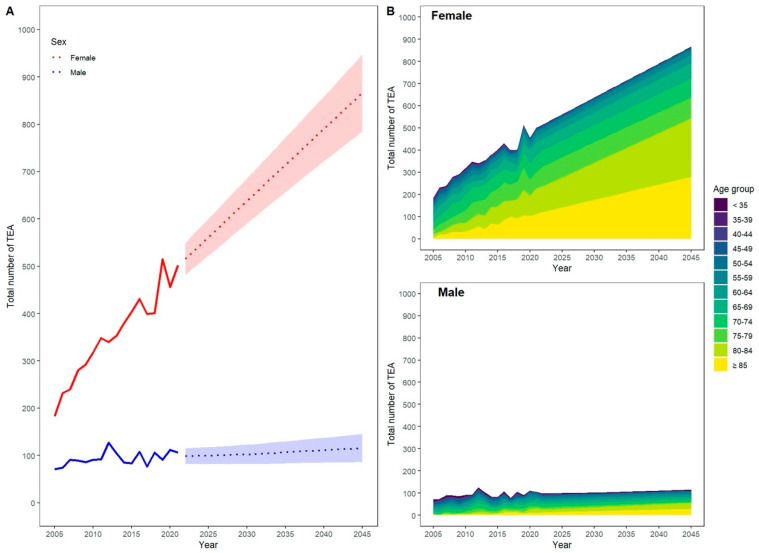
Increase in total elbow arthroplasty (TEA) by age, (**A**) total increase with CI, (**B**) total increase by sex and age group.

**Figure 4 healthcare-12-01322-f004:**
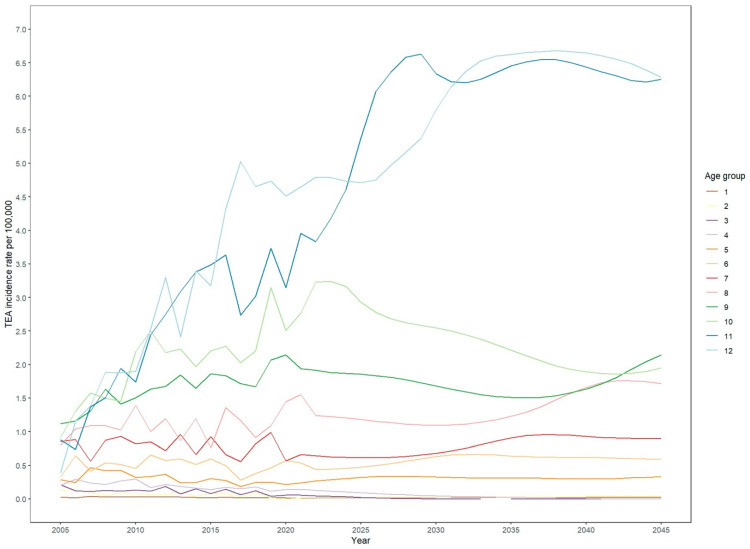
Development of implanted TEA incidence by age group.

**Table 1 healthcare-12-01322-t001:** Accuracy of predictions.

Model	MAPE	RMSE	MAE
ARIMA	7.30%	47.9	39.3
ETS	5.80%	37.4	31.6
Poisson	8.20%	58.8	51.4

Accuracy of the different prediction models used in this study. Mean absolute percentage error (MAPE), Root mean squared error (RMSE), and mean absolute error (MAE).

**Table 2 healthcare-12-01322-t002:** Predictions of numbers for the upcoming years.

Year	Absolute Numbers	Relative Increase [%]	Population	Incidence Rate (per 100,000)	Relative Increase [%]
2021	608		82,675,100	0.74	
2022	614 [580–648]	0.94	83,620,000	0.73 [0.69–0.77]	−0.20
2023	630 [594–665]	3.55	83,701,000	0.75 [0.71–0.79]	2.28
2024	645 [608–682]	6.14	83,752,000	0.77 [0.73–0.81]	4.77
2025	661 [622–700]	8.73	83,759,000	0.79 [0.74–0.84]	7.32
2026	677 [636–717]	11.31	83,749,000	0.81 [0.76–0.86]	9.88
2027	692 [650–735]	13.88	83,711,000	0.83 [0.78–0.88]	12.47
2028	708 [664–752]	16.47	83,683,000	0.85 [0.79–0.90]	15.06
2029	724 [678–770]	19.07	83,638,000	0.87 [0.81–0.92]	17.70
2030	740 [691–787]	21.68	83,572,000	0.89 [0.83–0.94]	20.37
2031	756 [705–805]	24.33	83,515,000	0.91 [0.84–0.96]	23.08
2032	772 [718–823]	27.00	83,443,000	0.93 [0.86–0.99]	25.83
2033	788 [731–840]	29.66	83,367,000	0.95 [0.88–1.01]	28.58
2034	804 [745–858]	32.31	83,290,000	0.97 [0.89–1.03]	31.34
2035	821 [758–876]	34.97	83,213,000	0.99 [0.91–1.05]	34.10
2036	837 [771–894]	37.62	83,125,000	1.01 [0.93–1.08]	36.88
2037	853 [783–912]	40.28	83,028,000	1.03 [0.94–1.10]	39.68
2038	869 [796–930]	42.92	82,925,000	1.05 [0.96–1.12]	42.49
2039	885 [809–948]	45.57	82,823,000	1.07 [0.98–1.15]	45.31
2040	901 [821–967]	48.22	82,707,000	1.09 [0.99–1.17]	48.16
2041	917 [834–985]	50.86	82,587,000	1.11 [1.01–1.19]	51.02
2042	933 [846–1004]	53.51	82,470,000	1.13 [1.03–1.22]	53.89
2043	949 [858–1023]	56.16	82,330,000	1.15 [1.04–1.24]	56.82
2044	966 [870–1041]	58.83	82,200,000	1.17 [1.06–1.27]	59.75
2045	982 [882–1060]	61.50	82,055,000	1.20 [1.07–1.29]	62.72

This table Predicted total numbers and incidence rates of TEA for projection years in relation to 2021. The corresponding 95% CIs are shown in parentheses. The incidence rate is given per 100,000 individuals, with a relative increase in % compared to the 2021 baseline data.

## Data Availability

The data used in this study are available upon request. Interested researchers can obtain access to the dataset by contacting the corresponding author. This ensures transparency and allows for the reproducibility and verification of the study’s findings.

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
