# Peer review of "The Future of Total Elbow Arthroplasty: A Statistical Forecast Model for Germany"

_healthcare, 2024, doi:10.3390/healthcare12131322_

Round 1

Reviewer 1 Report

Comments and Suggestions for Authors

The authors provide a statistical forecast for the development of total elbow arthroplasties (TEAs) in Germany until 2045, using an autoregressive integrated moving average (ARIMA), Error-Trend-Seasonality (ETS), and Poisson model to forecast trends in total elbow arthroplasty based on demographic information and official procedure statistics. The number of TEAs is projected to increase continuously from 2021 to 2045. This will result in a total number of 982 (TEAs) in 2045 of mostly elderly patients above 80 years. Female patients will receive TEAs 7.5 times more often than men. The topic has been handled appropriately and the references are appropriate.

Some comments:

Figs 1-3: Texts on the Y-axis must be put in a larger font. Now they are hard to read.

Fig. 4: Very difficult to read. The colors do not stand out. Would it be possible, for example, to put the right number at the right end of each curve? The age groups should also be put in this table as in Fig. 3.

line 136: The population is expected to increase from 82.7 million in 2021 to 83.9 million in 2022 and to continue to increase (?) annually until it reaches 82.1 million in 2045.

line 138: This increase in population is reflected in the projected total number of TEA procedures, which is expected to rise from 608 in 2021 to 982 in 2045. In fact, a decrease after 2037

line 144: After 2031, the incidence rate is expected to increase more quickly in females 144 than males (Fig.3). WHY?

Author Response

Comment 1: Figs 1-3: Texts on the Y-axis must be put in a larger font. Now they are hard to read.

Answer: I agree, changed.

Comment 2: Fig. 4: Very difficult to read. The colors do not stand out. Would it be possible, for example, to put the right number at the right end of each curve? The age groups should also be put in this table as in Fig. 3.

Answer: Indeed, this is a very complex figure. It contains a lot of data. I added the age groups to the end of each line to make it easier to find the matching age group/line. 

Comment 3: line 136: The population is expected to increase from 82.7 million in 2021 to 83.9 million in 2022 and to continue to increase (?) annually until it reaches 82.1 million in 2045.

Answer: Thank you, this is an incorrect statement. Please see my changes: 

"The population is expected to increase from 82.7 million in 2021 to 83.76 million in 2024 and will then slightly decline annually until it reaches 82.1 million in 2045. This increase in population, in combination with increasing cases and an overall ageing population, is reflected in the projected total number of TEA procedures, which is expected to rise from 608 in 2021 to 982 in 2045."

Comment 4: line 138: This increase in population is reflected in the projected total number of TEA procedures, which is expected to rise from 608 in 2021 to 982 in 2045. In fact, a decrease after 2037

Answer: See my changes above. The total number is expected to increase due to the development in the last years and overall ageing. 

Comment 5: line 144: After 2031, the incidence rate is expected to increase more quickly in females 144 than males (Fig.3). WHY?

Answer: This explanation is part of the discussion. Line 166: 

"This is further compounded by the fact that women generally have a more significant life expectancy than men, which partially explains the sex differences in projected cases for TEA"

Reviewer 2 Report

Comments and Suggestions for Authors

Comprehensive manuscript concerning statistical prediction models on TEA procedures in the years to come as we are facing an ageing and vital population.

The analyses are conducted thoroughly and explained as well as the results depicted clearly.

The presentation of the figure 4 and table 2 in my opinion are of lesser interest and could be left out. Figure 4 since it is abundantly described and rather obvious that in the young TEA is not indicated often and a rise in indication is this group is not expectated. A sentence describing this would suffice, rendering the possibility to take out the slightly chaotic figure.

As for table 2, the exact numbers per year with CI's might be of interest to the specialized statistical reader, though add little to the data presented and the conclusions drawn. Advice is to take this extensive table out.

What would be interesting on the other hand is to see in a figure how the TEA numbers develop concerning the main indication groups: autoimmune arthritis, traumatic (and post-traumatic). I would propose to put in a figure that depicts the descending curve for 'arthritic' TEA's as opposed to 'traumatic' TEA's.

Author Response

Comment 1: The presentation of the figure 4 and table 2 in my opinion are of lesser interest and could be left out. Figure 4 since it is abundantly described and rather obvious that in the young TEA is not indicated often and a rise in indication is this group is not expectated. A sentence describing this would suffice, rendering the possibility to take out the slightly chaotic figure. 

Answer: Thank you for your comment. I think this figure is a very complex one and contains a lot of data. It was very hard to put the development of the different age groups into a chart/ table to give a good overview. We added the age groups to the end of each line. We believe this makes it easier to understand. We also believe this figure helps to understand the numbers in the text. We would like to keep it in the paper. 

Comment 2: As for table 2, the exact numbers per year with CI's might be of interest to the specialized statistical reader, though add little to the data presented and the conclusions drawn. Advice is to take this extensive table out.

Answer: See above. 

Comment 3: What would be interesting on the other hand is to see in a figure how the TEA numbers develop concerning the main indication groups: autoimmune arthritis, traumatic (and post-traumatic). I would propose to put in a figure that depicts the descending curve for 'arthritic' TEA's as opposed to 'traumatic' TEA's

Answer: Thank you, this is a very good thought. Unfortunately, the official data given does not come with a code for diagnosis. That´s why we are unable to give statistical analysis about the indications for TEA. 

Reviewer 3 Report

Comments and Suggestions for Authors

The authors attempt to predict the use of total elbow arthroplasty in Germany in the next 20 years. This topic is quite interesting due to the expected changes in both demographics and indications for using TEA.

For my area of expertise, all information regarding total elbow arthroplasty is corrected and well-reported. The study design seems good. Conclusions are clear and supported by results. The only incoherence I found in the manuscript is the following:

In lines 137-138, you state population will increase, but the numbers you show are decreasing between 2022 and 2045.

Regarding the statistical part, I don't have the knowledge to evaluate and judge a so advanced statistic, so I would advise the editors to ask an expert in medical statistics to review this paper, too.

Author Response

Comment 1: In lines 137-138, you state population will increase, but the numbers you show are decreasing between 2022 and 2045.

Answer: 

Thank you, this is an incorrect statement. Please see my changes: 

"The population is expected to increase from 82.7 million in 2021 to 83.76 million in 2024 and will then slightly decline annually until it reaches 82.1 million in 2045. This increase in population, in combination with increasing cases and an overall ageing population, is reflected in the projected total number of TEA procedures, which is expected to rise from 608 in 2021 to 982 in 2045."